# Ambient temperature and mental health hospitalizations in Bern, Switzerland: A 45-year time-series study

Marvin Bundo[1,2,3], Evan de Schrijver[1,2,3], Andrea Federspiel[4], Andrea Toreti[5], Elena Xoplaki[6,7], Jürg Luterbacher[8], Oscar H. Franco[1], Thomas Müller[4,9]*, Ana M. Vicedo-Cabrera[1,2]*

1 Institute of Social and Preventive Medicine, University of Bern, Bern, Switzerland, 2 Oeschger Center for Climate Change Research, University of Bern, Bern, Switzerland, 3 Graduate School for Health Sciences, University of Bern, Bern, Switzerland, 4 Translational Research Center (TRC), University Hospital of Psychiatry and Psychotherapy University of Bern, Bern, Switzerland, 5 European Commission, Joint Research Centre, Ispra, Italy, 6 Department of Geography, Climatology, Climate Dynamics and Climate Change, Justus Liebig University Giessen, Giessen, Germany, 7 Center for International Development and Environmental Research (ZEU), Justus Liebig University Giessen, Giessen, Germany, 8 World Meteorological Organization (WMO), Science and Innovation Department, Geneva, Switzerland, 9 Privatclinic Meiringen, Meiringen, Switzerland

* anamaria.vicedo@ispm.unibe.ch (AMVC); thomas.mueller@upd.unibe.ch (TM)

**Data Availability Statement:** The data used in this study is held in an open data repository managed by the University of Bern (https://boris.unibe.ch/152745/).

## Abstract

### Background

Psychiatric disorders constitute a major public health concern that are associated with substantial health and socioeconomic burden. Psychiatric patients may be more vulnerable to high temperatures, which under current climate change projections will most likely increase the burden of this public health concern.

### Objective

This study investigated the short-term association between ambient temperature and mental health hospitalizations in Bern, Switzerland.

### Methods

Daily hospitalizations for mental disorders between 1973 and 2017 were collected from the University Hospital of Psychiatry and Psychotherapy in Bern. Population-weighted daily mean ambient temperatures were derived for the catchment area of the hospital from 2.3-km gridded weather maps. Conditional quasi-Poisson regression with distributed lag linear models were applied to assess the association up to three days after the exposure. Stratified analyses were conducted by age, sex, and subdiagnosis, and by subperiods (1973–1989 and 1990–2017). Additional subanalyses were performed to assess whether larger risks were found during the warm season or were due to heatwaves.

**Funding:** This project has received funding in the form of salaries for the authors MB and EDV from the European Union's Horizon 2020 research and innovation programme under the Marie Skłodowska-Curie grant agreement No 801076, through the SSPH+ Global PhD Fellowship Programme in Public Health Sciences (GlobalP3HS) of the Swiss School of Public Health. The author EX acknowledges support by the Academy of Athens and the Greek "National Research Network on Climate Change and its Impact" (project code 200/937) and the German Federal Ministry of Education and Research (BMBF) (01LR2002F) projects NUKLEUS and ClimXtreme (01LP1903C). Privatclinic Meiringen (Switzerland) provided support in the form of salaries for the author TM. None of the funding sources mentioned here have any additional role in the study design, data collection and analysis, decision to publish, or preparation of the manuscript. The specific roles of these authors are articulated in the 'author contributions' section.

**Competing interests:** The author TM is employed by Privatclinic Meiringen, Switzerland and the author AT is employed by the European Commission. This does not alter our adherence to PLOS ONE policies on sharing data and materials. The views here expressed are those of the authors and do not necessarily reflect an official position of the European Commission. The authors have declared that no competing interests exist.

## Results

The study included a total number of 88,996 hospitalizations. Overall, the hospitalization risk increased linearly by 4.0% (95% CI 2.0%, 7.0%) for every 10˚C increase in mean daily temperature. No evidence of a nonlinear association or larger risks during the warm season or heatwaves was found. Similar estimates were found across for all sex and age categories, and larger risks were found for hospitalizations related to developmental disorders (29.0%; 95% CI 9.0%, 54.0%), schizophrenia (10.0%; 95% CI 4.0%, 15.0%), and for the later rather than the earlier period (5.0%; 95% CI 2.0%, 8.0% vs. 2.0%; 95% CI -3.0%, 8.0%).

## Conclusions

Our findings suggest that increasing temperatures could negatively affect mental status in psychiatric patients. Specific public health policies are urgently needed to protect this vulnerable population from the effects of climate change.

## Background

With currently around 450 million people affected worldwide, mental disorders constitute a substantial health burden [1]. Mental disorders are among the leading causes of ill health and disability worldwide with a lifetime prevalence rate between 18.1% and 36.1% [1,2]. In Switzerland, neuropsychiatric diseases are responsible for 35.1% of the total burden of disease and account for 16% of the total health care costs [3,4].

A deeper understanding of the potential risk factors and mechanisms of mental disorders is needed to reduce their health, social, and economic burden. Recently high temperatures have recently been reported to negatively affect patients with mental disorders [5]. Medications that interfere with physiological homeostasis and/or increase in stress hormones are among prominent potential factors involved in mortality and morbidity related to heat exposure [6,7].

The impact of weather on mental health is gaining much attention in public health research lately, in particular in connection with the potential consequences that climate change may have on this vulnerable population in the future [8]. Several epidemiological investigations have recently showed an association between increased ambient temperatures and a higher risk for mental health outcomes such as emergency room visits, hospitalizations, and suicides [9–15]. Despite the relevance of the research topic, the evidence generated so far is limited, compared to other health outcomes (i.e. cardiovascular or respiratory mortality or morbidity) [16]. For example, these studies mostly focused on heatwaves or extreme heat or did not account for other weather or environmental factors. These studies have also examined a relatively short period or were restricted to a subsample of specific psychiatric disorders [5]. Furthermore, no studies have yet been conducted in Europe so far, and specifically in Switzerland, on the relationship between ambient temperatures and mental health hospitalizations. Only one recent study showed that warmer temperatures led to higher suicide rates in different regions, including Switzerland [14].

Now we face the challenges of global warming with increasing number of weather and climate extremes [17]. Climate change poses a serious threat to human health, including mental health [18]. Patients with mental disorders are one among the vulnerable groups that may be disproportionately affected from the effects of climate change [19]. The incidence and

prevalence of mental disorders are increasing and are projected to be one of the leading causes of disease burden worldwide by 2030 [20]. The impact of mental disorders on health care and the economic system is likely to be exacerbated by climate change. A better understanding of the links between climate and mental health in the 21st century is therefore crucial to provide policymakers and health professionals with evidence to inform policies targeted at the needs of mental health patients.

Here, we conducted a comprehensive assessment of the association between ambient temperatures and hospitalizations due to mental disorders in Bern, Switzerland, over a long period of 45 years. We assessed the association across population subgroups and specific subdiagnoses and explored potential temporal and seasonal patterns of the effects.

## Methods

### Hospitalization data collection

We collected anonymized individual data on all hospitalizations in the University Psychiatric Hospital in Bern, Switzerland for the period between 1973 and 2017. This hospital provides secondary and tertiary treatment in psychiatry for the city of Bern and 18 surrounding municipalities in the canton of Bern, serving a total population of 275,079 inhabitants. The catchment area of the hospital is presented in the map in Fig 1. Hospitalizations between 1973 and 2003 were classified according to the International Classification of Diseases 9 (ICD-9) and according to the ICD-10 for the period between 1993 and 2017. ICD-9 codes for specific subdiagnoses were converted to ICD-10 using a conversion function from the R package "icd". The conversion was done using General Equivalence Mappings (GEMs) proposed by the Centers for Medicare & Medicaid Services (CMS) [21]. Hospitalizations were then aggregated into total daily counts by date of hospitalization and by sex, age group (0–64, $\geq$65 years), and specific subdiagnoses (see the list of categories in Table 1).

### Meteorological and air pollution data

We used 1.6x2.3 km gridded daily mean temperatures provided by MeteoSwiss [22], and population data from EOSDIS (Earth Observing System Data and Information System, UN WPP-Adjusted Population Count, v4.11–2010) on a 1x1 km resolution to derive population-weighted daily mean temperatures for the catchment area of the hospital for the 45 consecutive years. This population-weighted temperature metric has been shown to be a better approximation of the average exposure of the population living in a large area with highly heterogeneous climate and orography [23]. By accounting for the differential population distribution, we are able to partially overcome potential exposure misclassification usually found when using data from a unique weather station. We used the ratio between the population residing in the corresponding grid cell and the total population residing in the catchment area to compute the weights of temperature cells.

Daily relative humidity, precipitation totals, atmospheric pressure, sunshine duration and wind speed from the Bern/Zollikofen weather station, located in a suburb 7.4 km north of the city of Bern, since gridded products for these variables were not available. Data were retrieved from the IDAWEB database (MeteoSwiss) for the study period. We defined atmospheric pressure difference as the difference in pressure between the current and the previous day.

The Office of Environment and Energy of the Economic, Energy, and Environmental Directorate of the Canton of Bern provided air pollution data. Daily average levels of particulate matter with diameter less than 10μm (PM10), nitrogen dioxide (NO$_2$), and ozone (O$_3$) were derived from hourly measurements registered by the Bern-Bollwerk station (located near

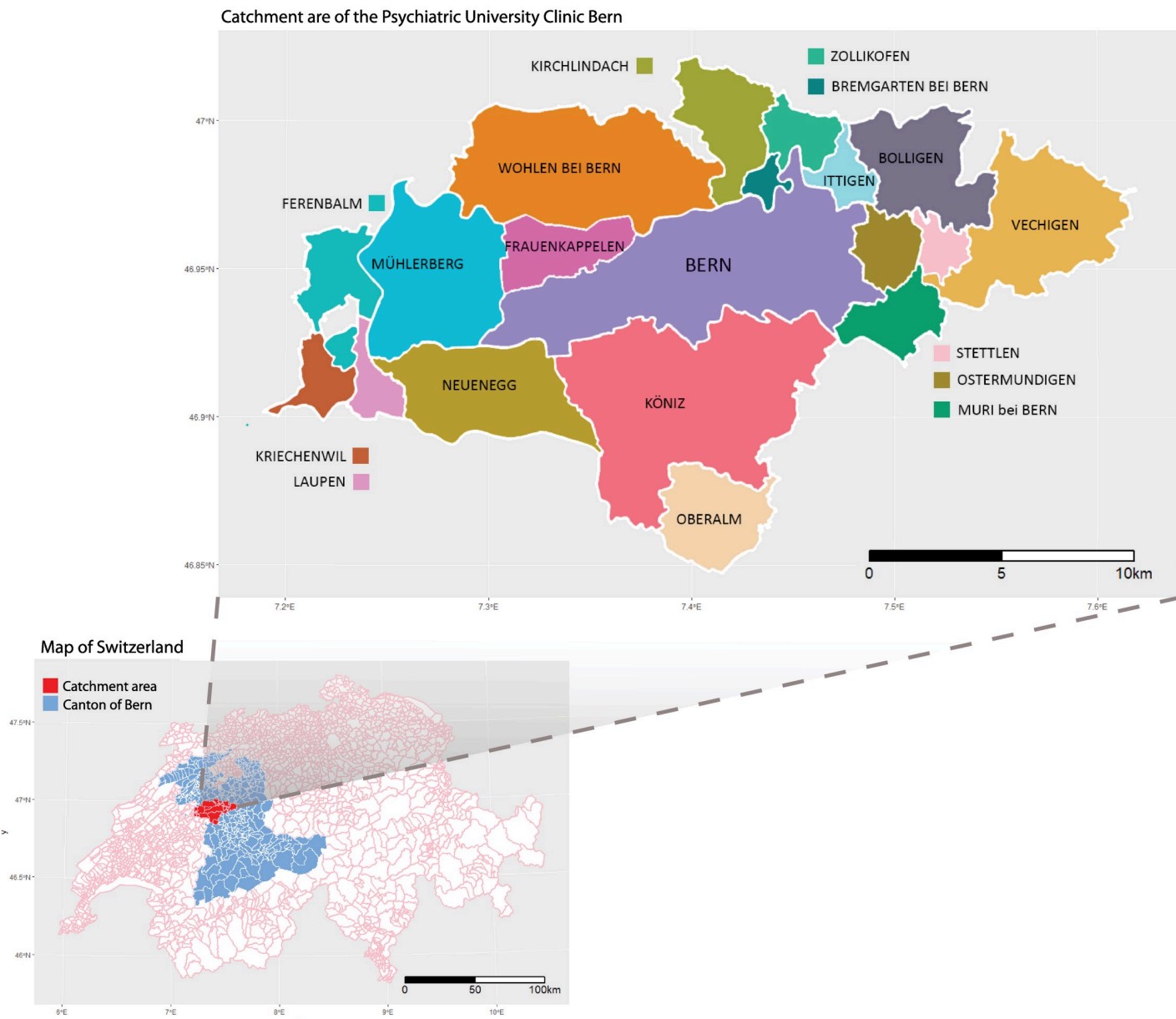

**Fig 1. Catchment area of the University Psychiatric Hospital, Bern, Switzerland.**

the city center of Bern) of the NABEL network (National Air Pollution Monitoring Network). These data were only available for the period 1991–2017.

## Statistical analysis

To assess the short-term association between ambient temperature and hospitalizations due to mental disorders we conducted an aggregated time-stratified case-crossover analysis [24,25]. This is a type of case-crossover design in which a common exposure level is defined for all subjects of the study area. Thus, the data retain the usual time-series structure, with the outcome

**Table 1. Descriptive statistics for mental disorder hospitalizations in University Psychiatric Hospital Bern (Switzerland), 1973–2017.**

| | | N | % |
|---|---|---|---|
| Total | | 88996 | |
| Sex | Male | 44370 | 49.9 |
| | Female | 44626 | 50.1 |
| Age | <65 years old | 75400 | 84.7 |
| | ≥65 years old | 13596 | 15.3 |
| Diagnosis | Total* | 83549* | |
| | Organic, including symptomatic, mental disorders (F00-F09) | 4211 | 5.0 |
| | Mental and behavioral disorders due to psychoactive substance use (F10-F19) | 19000 | 22.7 |
| | Schizophrenia, schizotypal and delusional disorders (F20-29) | 22447 | 26.9 |
| | Mood disorders (F30-39) | 17052 | 20.4 |
| | Neurotic disorders (F40-59) | 12096 | 14.5 |
| | Developmental disorders (F80-98) | 2147 | 2.6 |
| | Disorders of adult personality and behavior (F60-69) | 5863 | 7.0 |
| | Mental retardation (F70-79) | 733 | 0.9 |
| Subperiods | 1973–1989 | 18567 | 21.0 |
| | 1990–2017 | 70429 | 79.0 |

* Data on diagnosis not available for 5447 patients (6.5%). Corresponding IDC-10 code in brackets.

and exposure respectively defined as daily counts and levels. The model assumes stationarity in the association and contrary to the usual time-series analysis. This approach applies the idea of the bidirectional, time-stratified case-crossover design to control for long-term and seasonal trends by matching case and control days within year, month, and day of the week. For example, each Monday in September 1991 would be compared to all other Mondays of the month and year. Thus, any temporal patterns occurring at longer time scales beyond this timeframe are removed by design (i.e. long-term trends, seasonality) [26]. This implies that not accounting for external factors slowly changing in time or altering temporal patterns at longer time scale (i.e. changes in diagnostic criteria) would not introduce any bias in our estimates. This design has been shown to be more efficient when the number of events per day is low and less computationally intense than the individual case-crossover analysis, and it allows for controlling for overdispersion and autocorrelation [24].

We used generalized linear models with conditional quasi-Poisson regression accounting for overdispersion with the daily number of mental health hospitalizations as dependent variable [24,25]. Each day with events was matched with its control days according to the year, month, and day of the week by including a stratum variable in the model. As previous studies have shown, the short-term effect of temperature on health can be not linear with increased risks in heat and cold ranges, and also not limited to the same day of exposure with effects still persisting for several days [9,10,12,27,28]. Although the current literature on mental health hospitalizations and temperature suggests a potential linear association, we chose to explore potential non-linearities using distributed lag non-linear models, a flexible framework that allows us to compare different parametrizations of the exposure-response association and account for lagged dependencies [25]. The selection of the model specifications was based on quasi-Akaike information criteria across a range of functions defined in the so-called cross-basis term of temperature (S1 Methods Appendix). Briefly, these included linear and nonlinear functions (quadratic b-spline) in the exposure-response dimension, and different functions modelling the lag-response dimension with a maximum lag of 3 and 7 days. Then, according

to qAIC criterion, the final model consisted of unconstrained distributed lag linear model (i.e. assuming a linear association) with a maximum lag of three days [29]. Additionally, other weather variables (humidity, precipitation, atmospheric pressure, atmospheric pressure difference, sunshine duration, wind speed) and air pollutants ($NO_2$, PM10, and $O_3$) were considered as potential confounders of the temperature-hospitalization association as previously described [9,10,12]. Confounding was assessed by separately including each of these environmental factors in the main model with temperature and was assessed whether the association between temperature and hospitalizations changed or remained similar (see S2 Methods Appendix for more details on the method and results). Based on this exploratory analysis, none of the considered variables was shown to confound the temperature-hospitalization association, thus the main model did not include any of these environmental factors.

The main analysis was performed for total daily counts and by subgroups of sex, age (patients aged 0–64 and $\geq$ 65 years old), and ICD-10 classification: organic mental disorders (F00-F09), psychoactive substance use (F10-F19), schizophrenia (F20-29), mood disorders (F30-39), neurotic disorders (F40-59), adult and behavioral personality disorders (F60-69), mental retardation (F70-79), and developmental disorder (F80-98). A temporal analysis was conducted to assess whether the association (overall and by subgroup) differed between recent years (1990 to 2017) and previous decades (1973 to 1989). Results were reported as the percentage change in risk per 10˚C increase in daily mean temperature on the same day of the exposure and the following three days (i.e. the overall cumulative risk from 0 (day of exposure) to lag 3) with the corresponding 95% confidence intervals (CI).

An additional set of subanalyses was performed to assess whether the association was larger when only high temperatures or extreme heat events were considered. Therefore, a seasonal analysis was first performed with the same model specifications, but restricting the study period to the warm season corresponding to the four consecutive warmest months (May to September). For extreme heat events, heatwaves were defined as events during the warm season (May-September) when for at least for two/three consecutive days temperatures were equal or above the 90th and 95th percentile temperature thresholds. The association with extreme heat events was modelled using a binary variable and the same modelling strategy as in the main analysis (which accounted for effects delayed up to three days). Additional subanalyses were also conducted for other definitions of heatwaves (92.5th or 97.5th percentile of temperature from May to September). As there are no standard definitions for heatwaves, these definitions were taken from previous similar studies [30].

All the statistical analyses were conducted in R (version 3.4.4, R Development Core Team).

## Results

### Descriptive statistics

Between 1 January 1973 and 31 December 2017, there were a total of 88,996 hospitalizations due to mental disorders at the University Psychiatric Hospital in Bern. Among these, 50% of the patients were male and 15% aged 65 or above (Table 1). The most frequent hospitalization subdiagnoses were schizophrenia (27%), mental and behavioral disorders due to psychoactive substance use (23%), and mood disorders (20%). The median daily temperature for the catchment area was 8.7˚C (interquartile range (IQR): 2.7–14.6) for the entire study period, 7.9˚C (IQR: 2.1–14.0) for the period 1973–1989, and 9.2˚C (IQR: 3.1–15.0) for the period 1990–2017 (Table 2). The descriptive statistics for the other weather variables, heatwave events for different definitions (temperatures equal or above the 92.5th and 97.5th percentile and duration of at least two or three days), and air pollutants are shown in S1 Table. Fig 2 shows that both daily counts of mental health disorders hospitalizations and mean temperature increased between

**Table 2. Descriptive statistics for daily mean ambient temperature (for the entire study period, the warm season (May-September) and each subperiod) and heat-waves (two or three days of duration and temperature equal or above the 90ᵗʰ or 95ᵗʰ percentile.**

| Daily Mean Ambient temperature (˚C) | Minimum | 25th Pct | Median | Mean | 75th Pct | Maximum |
|---|---|---|---|---|---|---|
| Whole year, 1973–2017 | -18.0 | 2.7 | 8.7 | 8.6 | 14.6 | 27.4 |
| Warm season (May-September), 1973–2017 | 1.7 | 12.7 | 18.2 | 18.4 | 15.5 | 27.4 |
| Whole year, subperiod 1973–1989 | -18.0 | 2.1 | 7.9 | 7.9 | 14.0 | 25.3 |
| Whole year, subperiod 1990–2017 | -14. | 3.1 | 9.2 | 9.0 | 15.0 | 27.4 |
| **Extreme heat events (May-September)** | | | **Number of heatwave days** | **%** | **Average nr. of heatwave days per year** | |
| > = 2-day duration heatwave with temperature ≥ 90ᵗʰ > = percentile (18.3˚C) | | | 1558 | 22.6 | 34.6 | |
| > = 2-day duration heatwave with temperature ≥ 95ᵗʰ percentile (20.1˚C) | | | 756 | 11.0 | 16.8 | |
| > = 3-day duration heatwave with temperature ≥ 90ᵗʰ percentile (18.3˚C) | | | 1424 | 20.7 | 31.6 | |
| > = 3-day duration heatwave with temperature ≥ 95ᵗʰ percentile (20.1˚C) | | | 628 | 9.1 | 14.0 | |

1973 and 2017. Deviation from monotonicity in the long-term behaviour and change points can be also observed (e.g. steep increase from 1996 onwards). These could be possibly attributed to changes in diagnostic criteria or administrative decisions (e.g. reduction in the length of the stay which would allow for the admission of more patients). However, as mentioned in the method section, not accounting for these factors altering long-term temporal patterns would not introduce bias in the association estimates, as its potential confounding effect is removed by design.

## Association between ambient temperature and mental health hospitalizations

As shown in Fig 3, the risk of hospitalizations due to mental disorders increased linearly by 4.0% (RR: 1.04: 95% CI 1.02, 1.07) for every 10˚C increase in daily mean temperature. In subgroup analysis, similar estimates of association were found across sex, with 5.0% in men (RR: 1.05: 95% CI 1.00, 1.09) and 4.0% in women (RR: 1.04: 95% CI 1.00, 1.08), and age categories with 4.0% (RR: 1.04: 95% CI 0.97, 1.11) ≥65 years old and 4.0% (RR: 1.04: 95% CI 1.01,

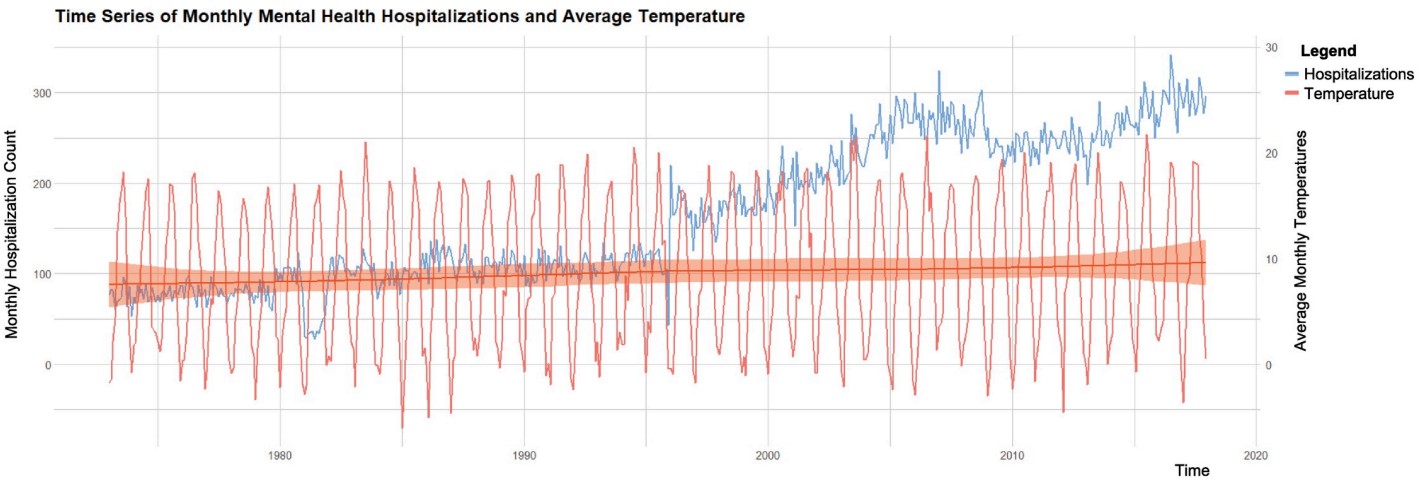

**Fig 2. Temporal evolution of the hospitalizations due to mental disorders and mean temperature across the 45-year study period (1973–2017).**

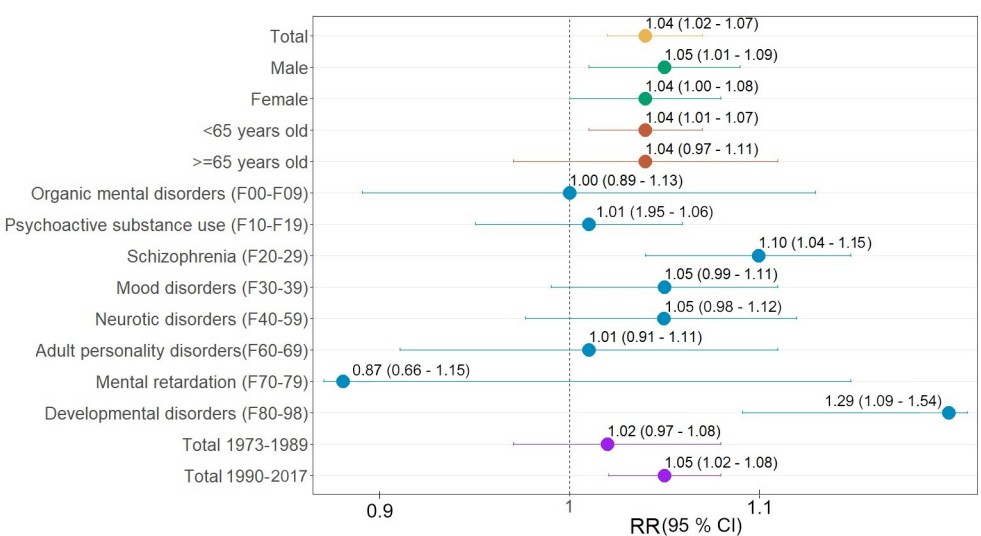

**Fig 3. Relative risks (RR, 95% confidence interval (CI)) of mental disorders hospitalizations per 10˚C increase in mean daily temperature (lag 03).** Overall result, subgroup analysis by age, sex and subdiagnosis, and temporal analysis. Null hypothesis is that there is no association (RR = 1), thus one can reject the null hypothesis when 95% confidence interval does not include 1.

1.07) <65 years old). By subdiagnoses groups, larger association estimates were found for developmental disorders (F80-98) (29.0%; RR: 1.29: 95% CI 1.09, 1.54) and schizophrenia (F20-29) (10.0%; RR: 1.10: 95% CI 1.04, 1.15). Positive associations, although less precise, were also found for mental disorders due to psychoactive substance use (F10-F19), mood disorders (F30-39), neurotic disorders (F40-59), and adult personality disorders (F60-69). No evidence of an association between temperature and the other subdiagnoses (organic mental disorders (F00-F09) and mental retardation (F70-79)) was found.

The overall cumulative association estimate was slightly larger in the more recent period (1990–2017) of the study, 5.0% (1.05: 95% CI 1.02, 1.08), compared to the first period (1973–1989), 2.0% (RR: 1.02: 95% CI 0.97, 1.08) (Fig 3). As shown in S2 Table, this pattern was repeated for these two periods among men (6.0%; RR: 1.06: 95% CI 1.02, 1.11, vs. 0.0%; RR: 1.00: 95% CI 0.93, 1.09), and among those <65 years old (6.0%; RR: 1.06: 95% CI 1.02, 1.09 vs. 0.0%; RR: 1.00: 95% CI 0.94, 1.06), as well as for diagnoses of schizophrenia (10.0%; RR: 1.10: 95% CI 1.04, 1.17, vs. 6.0%; RR: 1.07: 95% CI 0.95, 1.18) and developmental disorders (31.0%; RR: 1.31: 95% CI 1.08, 1.60, vs. 15.0%; RR: 1.14: 95% CI 0.77, 1.73).

When limited to the warm season (May-September), the overall cumulative association remained similar and was somewhat less precise compared to the whole-year estimate (Fig 4). Larger risk of hospitalization, although unprecise, was found for heatwaves. In particular, a 12.0% increase in risk (RR: 1.12: CI 0.97, 1.30) was estimated for the most extreme episodes of three consecutive days with temperatures above the 95th percentile. Similar estimates were found for heatwaves of different duration, which suggests no additional risk attributed to the heatwave length. Similar patterns were found for heatwaves based on the 92.5th and 97.5th percentiles (S3 Table).

The association estimates were found to be robust when controlling for air pollutants and other meteorological factors (S1 and S2 Figs) and when testing for different model specifications (S3 and S4 Figs, and S4 Table). Notably, as shown in S3 Fig, the association was observable until three days after exposure (lag 3) that would justify the choice of accounting until 3 days of lag in the main analysis. Additionally, the model assuming a non-linear relationship

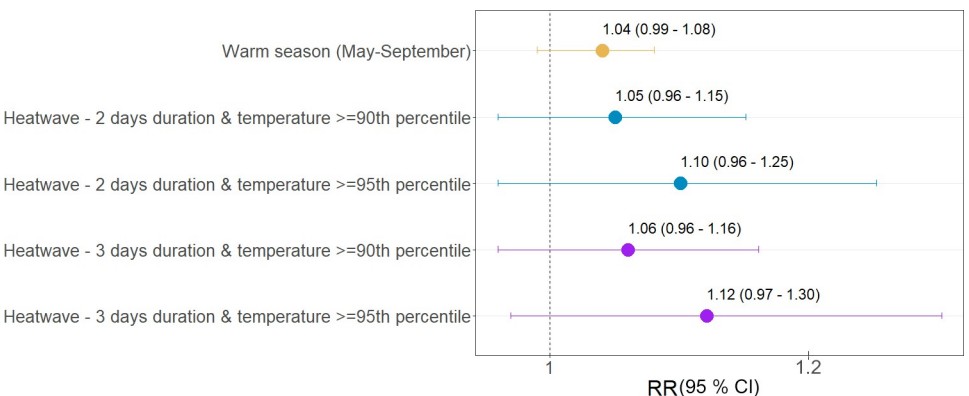

**Fig 4. Relative risks (RR, 95% confidence interval) of mental disorders hospitalizations per 10 C increase in daily mean temperature during the warm season (May-September) and heatwaves.** Null hypothesis is that there is no association (RR = 1), thus one can reject the null hypothesis when 95% confidence interval does not include 1.

did not provide a better fit and the resulting curve was close to linear (S4 Fig), suggesting no increased risk is associated to cold temperatures.

## Discussion

The risk of mental health hospitalizations increased by 4% per each 10°C increase in ambient temperature in Bern between 1973 and 2017. These findings are consistent with the conclusions of similar studies conducted elsewhere [9,10,31] that assess mental health outcomes such as emergency room visits [12] and/or mortality due to mental disorders [11]. The association we estimated was linear, as it was in studies conducted in Toronto and Hong Kong [9,31], with no evidence of yet larger risks attributed to warmer temperatures, whether during the warm season or heatwaves. Another study has also shown a linear relationship between temperature and suicide in Switzerland and other Western countries [32]. Two previous studies showed an increased risk of mental health hospitalizations during heatwaves, but it should be noted that these were conducted in Australia and Vietnam—two countries with completely different climates compared to Switzerland (all-year moderate climate in Switzerland compared to a Mediterranean climate, with warm, dry summers and mild winters in South Australia and hot summers and high humidity in Vietnam) [11,33]. In this study, only a weak evidence of larger risks was found for more extreme heatwaves (i.e. above 95th percentile) and, interestingly, the severity of the heatwave in terms of duration did not constitute additional risk, as suggested in previous studies [34]. The uncertainty of these estimates can be explained by the low statistical power as heatwaves are rare events in Switzerland, with an average of 14–35 heatwave days per year (depending on the definition of heatwaves) (Table 2) [35]. Finally, our results are also consistent with findings from previous studies suggesting that cold is not associated with an increase in mental health hospitalizations [9,10,31].

Several plausible mechanisms have been proposed to explain how ambient temperatures can affect mental health. For example, heat stress can affect psychophysiological functions by altering the levels of neurotransmitters like serotonin and dopamine (responsible for both thermoregulation and behavioural and mental states), and by disrupting the function of central and peripheral thermoregulation in the body [6,36–38]. Psychopharmacological drugs also contribute to altering thermoregulatory mechanisms such as sweating in mental health patients [39]. High temperatures can lead to central nervous system abnormalities (impaired cognitive function, altered execution of effective behavioural responses, altered short- and

long-term memory) [6]. The impaired cognitive function could make these individuals less able to consciously perceive environmental hazards and fail to protect themselves from heat exposure for prolonged periods [40,41]. Heat-induced sleep deprivation could also contribute to the exacerbation and maintenance of existing mental health symptoms [6]. Heat can cause psychological distress in mental health patients and lead to alcohol and substance abuse or aggressive behaviour [42]. Additionally, it should be noted that people who commit suicide, more frequently in patients with mental disorders, are reported to have decreased levels of L-tryptophan in their plasm, a biochemical which is also decreased by high temperatures [43]. The potential causal effects of ambient temperatures on mental health may be further clarified by Mendelian randomization analysis [44–48], due to the lack of the possibility of conduct randomized controlled trials.

The subgroup analysis showed that rising temperatures would equally affect men and women across all age groups. Previous studies offer mixed conclusions regarding the role of demographic factors or individual characteristics in the temperature-mental health outcomes association. For example, studies in China and Vietnam found a stronger association in the elderly [9,33], while other studies in Australia and Canada found larger effects among younger populations [31,49]. In addition, several studies have shown that women are generally more vulnerable to heat due to, for example, reduced sweating capacity during menopause, or to certain socioeconomic conditions (living alone and lower income) [50], whereas others have suggested that men are particularly vulnerable particularly to extreme heat events as they tend to have more outdoor work [51]. These contradictory findings could be explained by the different profiles of these subpopulations in terms of differential access to health care, which is determined by socioeconomic status, as well as individual characteristics such as family history, which could indirectly define the susceptibility of the patients within and between populations.

Patients with developmental disorders and schizophrenia seemed to be most affected by high temperatures in this study. Only one study so far has demonstrated an effect of heatwaves on psychological development [11]. This finding can be explained, for example, due to the fact that many patients with developmental disorders (e.g., autism) can be hyposensitive to external environmental hazards such as extreme temperatures [52]. Leading models of sensory hyper-reactivity postulate that due to elevated perceptual threshold individuals with autism spectrum disorder exhibit indifference (do not react or take measures against) to external stimuli such as pain and temperature [53–57]. Additionally, in patients with autism spectrum conditions, information processing can be altered, and therefore the reception of hazardous weather information (such as heatwaves) can be inhibited [58]. In individuals with attention-deficit/hyperactivity disorder the executive function, which is already deficient, can be further burdened by heat exposure [6]. Heat has also been shown to exacerbate tic symptoms in Tourette syndrome via normal heat-loss mechanisms through dopaminergic pathways [59]. Furthermore, heat has been shown to inhibit learning in schools and could, therefore, worsen further the condition of children with learning disabilities [60].

In relation to schizophrenic disorders, several studies showed positive associations between ambient temperature and increased incidence, disease symptomatology, hospitalization rate and/or acute exacerbations [5,11,61–64]. Patients with schizophrenia seem to have an impaired ability to compensate to heat stress due to a combination of peripheral ("impaired heat loss through peripheral vasodilatation via abnormalities in niacin and PGE1") and central mechanisms of thermoregulation ("disruption of the mesolimbic dopamine system") [65]. Other studies also suggest that in schizophrenic patients on medications, the inability to convey heat from the body's core to the periphery could be also a neuroleptic-induced side effect [65]. Heat stress has been shown also to alter the mental status and lead to delusional disorders,

which are not considered a separate category in ICD-10 but included in the category F20–F29, therefore contributing to the increased risk for this group [66,67]. Additionally, these patients are more prone to heat-related mortality due to accompanying chronic diseases, poor general health, and low socioeconomic status [68,69]. Inability to initiate adaptive behaviors against heat (such as fluid intake or adequate clothing), inability to take care of themselves, and a high degree of dependence on other persons are also important risk factors that make individuals with schizophrenia, but also with developmental disorders, highly susceptible to heat stress [6,11,66,67].

Previous studies have found increased risks of hospitalization due to mood disorders, neurotic disorders, mental retardation, and psychosis associated with heatwaves and/or high temperatures [5]. However, no clear evidence of an association was found in this study for these subdiagnoses, possibly due to the limited statistical power in these categories.

Our results show that the impact of high temperatures on mental health has increased in recent years. This may not be explained by the ageing of the population as previously reported [9,10,33], as age does not seem to influence the relationship between temperature and mental health hospitalizations. The increase in the prevalence of developmental disorders (one of the subdiagnoses most affected by increasing ambient temperatures) in recent years could be one possible explanation for this [70]. These findings would suggest that mental health patients in Switzerland may be more vulnerable to a warming climate in the future. Additionally, the larger risk in the more recent period would suggest that the improvement of infrastructure such as housing insulation and increased air conditioning, and also of the healthcare system and/or implementation of public health strategies that can include heat-health warning systems in recent decades may not have sufficiently mitigated the impact of increasing temperatures on mental health patients [71].

To our knowledge, this study represents the first comprehensive assessment of the impact of ambient temperature on mental health hospitalizations in Switzerland and Europe. The analysis applied state-of-the-art methodologies recently developed in climate change epidemiology to assess the ambient temperature–health outcomes association by considering different definitions of temperature (i.e. whole temperature range extreme heat events), study periods (i.e. whole year, warm season, later or recent periods) potential non-linear and delayed effects and characteristics of the patients and sub-diagnoses. Data from one of the main psychiatric hospitals in this country were used, including 88,996 hospitalizations across 45 years of time. The results of the sensitivity analyses suggest that our findings are robust to the modelling choices in terms of selection of the lags and functions to model the exposure-lag-response association. We applied aggregated case-crossover design with time-stratified matching approach which has been shown to properly control for long term and seasonal trends. It should be noted that confounding by time-invariant individual factors is removed by design [24]. Additionally, we found that temperature had an effect independent from the other environmental factors tested in the study (air pollutants or other meteorological factors) tested in the study. This is the first study in which the effect of temperature is controlled by all other meteorological factors. In a similar earlier study, only the effects of relative humidity and solar radiation were considered [12].

Limitations of this study include the assumption that exposure to ambient temperatures was the same across the entire study population, although this would only affect the precision of the estimates (i.e Berkson error) [72], though. However, it should be noted that population-weighted daily mean temperatures were used as a way to partially overcome potential exposure misclassification due to the large differences in population distribution across the catchment area derived from the irregular orography [23]. Second, this study did not assess differential effects by specific characteristics of the individuals, such as patients' socioeconomic status,

their current health conditions (i.e. potential comorbidities), health history and outpatient data (e.g. patients who did not require hospitalizations or emergency visits). Additionally, we could not explore relevant clinical characteristics of the admission such as the length of stay, the severity of symptoms, degree of impairment of social and occupational functioning, psychiatric drugs administered or whether it was the first admission or a recurrent case. Information n the latter would help clarifying if under medical advice patients are less vulnerable to heat. Third, some misclassification across subdiagnoses could be present, since several psychiatric disorders can have similar clinical symptoms. Fourth, residual confounding due to unknown factors changing on the same time scale of the association cannot be ruled out. Fifth, as shown in Fig 2 trends in hospitalizations displayed irregular temporal patterns. For example, we observe a steep increase after 1994, which could be due to changes in the diagnostic criteria or administrative reasons (e.g. shorten length of stay). However, it should be noted that these would have not affected our estimates nor introduced bias in our estimations, as confounding by long term trends and seasonal patterns are controlled for by design. Finally, although the study included data from one of the main psychiatric hospitals in Switzerland, the catchment area covered a limited geographical region of the country. Associations between temperature and health can be highly heterogeneous across locations even within countries and regions [14,27]. The conclusions of this study are therefore not necessarily representative of the Swiss population, or that of Europe. A well-designed trans-ethnic meta-analysis may help gain enough statistical power to clarify the association between ambient temperature and mental health hospitalizations with stratified analyses, according to ethnicity, sex, age, region, economic status, sub-diagnosis, and family history. And this strategy has already been successfully employed to clarify the potential associations between some genetic and environmental factors and risks of mental disorders and other health conditions [73–82].

## Conclusions

Overall we found that increasing ambient temperature could result in an increased risk for mental health hospitalizations in Bern. Findings also suggest that this association was mostly found in the last decades, despite the recent implementation of public health strategies against the effects of climate change. It is expected that the burden of mental health disorders could increase in the future as an indirect effect of climate change, for example due to migration of the populations and conflicts [83]. However, this study demonstrates that rising temperatures are having an effect on mental health—even in temperate climates. In the absence of interventions targeted at vulnerable populations, the direct impact upon mental health could be further amplified. This supports policy-makers developing specific public health interventions to counteract the negative health impacts of climate change and its human, social, and financial consequences. This study also paves the way for larger-scale studies on the association between temperature and mental health at national and international levels to gain a better understanding of the underlying pathophysiology of temperature's effect on mental health.

## Supporting information

**S1 Fig. Association estimates (relative risk (RR) and 95% confidence interval) of mental disorder hospitalizations for each meteorological factor (with and without control for temperature) and temperature controlled by meteorological factor.** Null hypothesis is that there is no association (RR = 1), thus one can reject the null hypothesis when 95% confidence interval does not include 1.
(TIF)

**S2 Fig. Association estimates (relative risk (RR) and 95% confidence interval (CI)) of mental disorders hospitalizations for each air pollutant (with and without control for temperature) and temperature controlled by air pollutant for the period 1991–2017.** Null hypothesis is that there is no association (RR = 1), thus one can reject the null hypothesis when 95% confidence interval does not include 1.
(TIF)

**S3 Fig. Sensitivity analysis: Association estimates (relative risk and 95% confidence interval) of mental disorder hospitalizations for 10˚C increase in mean daily temperature (lag 07).**
(TIF)

**S4 Fig. Sensitivity analysis: Comparison of the linear vs. non-linear exposure-response (quadratic b-spline with knots at 50th and 90th percentile) of the temperature–hospitalizations associations for the entire study period.**
(TIF)

**S1 Table. Descriptive statistics for other environmental factors and extreme heat events according to additional definitions used in the sensitivity analysis (event of two or three days of duration and temperature equal or above the 92.5th or 97.5th percentile during the warm season May-September).**
(DOCX)

**S2 Table. Relative risks (RR, 95% confidence interval (CI)) of mental disorders hospitalizations per 10˚C increase in daily mean temperature (lag 03)–Overall result, subgroup analysis by age, sex and diagnosis for two superiods: 1973–1989 and 1990–2017.**
(DOCX)

**S3 Table. Sensitivity analysis—Association estimates (relative risk (RR) and 95% confidence interval (CI)) using additional definitions for extreme heat events (event two and three days of duration and temperature equal or above the 92.5th or 97.5th percentile during warm season May-September).** Null hypothesis is that there is no association (RR = 1), thus one can reject the null hypothesis when 95% confidence interval does not include 1.
(DOCX)

**S4 Table. Sensitivity analysis–Association estimates (relative risk (RR) and 95% confidence interval (CI)) obtained in models using different specifications in the cross-basis of ambient temperature (whole year period).** Null hypothesis is that there is no association (RR = 1), thus one can reject the null hypothesis when 95% confidence interval does not include 1.
(DOCX)

**S1 Methods Appendix. Sensitivity analysis: Association estimates obtained from models using different cross-basis parameter of ambient temperature.**
(DOCX)

**S2 Methods Appendix. Selection of potential confounders: Other meteorological factors and air pollutants.**
(DOCX)

## Acknowledgments

We thank MeteoSwiss for providing the meteorological data. We also thank the Office of Environment and Energy of the Economic, Energy, and Environmental Directorate of the Canton of Bern for providing the air pollution data.

## Author Contributions

**Conceptualization:** Marvin Bundo, Andrea Toreti, Elena Xoplaki, Jürg Luterbacher, Oscar H. Franco, Thomas Müller, Ana M. Vicedo-Cabrera.

**Data curation:** Marvin Bundo, Evan de Schrijver, Andrea Federspiel, Elena Xoplaki, Jürg Luterbacher, Thomas Müller, Ana M. Vicedo-Cabrera.

**Formal analysis:** Marvin Bundo, Ana M. Vicedo-Cabrera.

**Investigation:** Marvin Bundo, Thomas Müller, Ana M. Vicedo-Cabrera.

**Methodology:** Marvin Bundo, Andrea Toreti, Thomas Müller, Ana M. Vicedo-Cabrera.

**Project administration:** Ana M. Vicedo-Cabrera.

**Supervision:** Oscar H. Franco, Thomas Müller, Ana M. Vicedo-Cabrera.

**Validation:** Ana M. Vicedo-Cabrera.

**Visualization:** Marvin Bundo, Evan de Schrijver.

**Writing – original draft:** Marvin Bundo, Ana M. Vicedo-Cabrera.

**Writing – review & editing:** Marvin Bundo, Evan de Schrijver, Andrea Federspiel, Andrea Toreti, Elena Xoplaki, Jürg Luterbacher, Oscar H. Franco, Thomas Müller, Ana M. Vicedo-Cabrera.

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
