## [Decision Letter · Decision Letter 0]

7 Dec 2020

PONE-D-20-33096

Ambient temperature and mental health hospitalizations in Bern, Switzerland: A 45-year time-series study

PLOS ONE

Dear Dr. Vicedo-Cabrera,

Thank you for submitting your manuscript to PLOS ONE. After careful consideration, we feel that it has merit but does not fully meet PLOS ONE’s publication criteria as it currently stands. Therefore, we invite you to submit a revised version of the manuscript that addresses the points raised during the review process.

We look forward to receiving your revised manuscript.

Kind regards,

Chin-Kuo Chang, Ph.D.

Academic Editor

PLOS ONE

Journal Requirements:

2.) Thank you for stating the following in the Financial Disclosure section:

'The authors received no specific funding for this work.'

We note that one or more of the authors are employed by a commercial company: Privatclinic Meiringen, Switzerland.

2.1. Please provide an amended Funding Statement declaring this commercial affiliation, as well as a statement regarding the Role of Funders in your study. If the funding organization did not play a role in the study design, data collection and analysis, decision to publish, or preparation of the manuscript and only provided financial support in the form of authors' salaries and/or research materials, please review your statements relating to the author contributions, and ensure you have specifically and accurately indicated the role(s) that these authors had in your study. You can update author roles in the Author Contributions section of the online submission form.

2.2. Please also provide an updated Competing Interests Statement declaring this commercial affiliation along with any other relevant declarations relating to employment, consultancy, patents, products in development, or marketed products, etc.  

3.) We note that you have indicated that data from this study are available upon request. PLOS only allows data to be available upon request if there are legal or ethical restrictions on sharing data publicly. For information on unacceptable data access restrictions, please see http://journals.plos.org/plosone/s/data-availability#loc-unacceptable-data-access-restrictions.

4.) We note that Figure 1 in your submission contain map images which may be copyrighted. All PLOS content is published under the Creative Commons Attribution License (CC BY 4.0), which means that the manuscript, images, and Supporting Information files will be freely available online, and any third party is permitted to access, download, copy, distribute, and use these materials in any way, even commercially, with proper attribution. For these reasons, we cannot publish previously copyrighted maps or satellite images created using proprietary data, such as Google software (Google Maps, Street View, and Earth). For more information, see our copyright guidelines: http://journals.plos.org/plosone/s/licenses-and-copyright.

4.1. You may seek permission from the original copyright holder of Figure1 to publish the content specifically under the CC BY 4.0 license. 

4.1. If you are unable to obtain permission from the original copyright holder to publish these figures under the CC BY 4.0 license or if the copyright holder’s requirements are incompatible with the CC BY 4.0 license, please either i) remove the figure or ii) supply a replacement figure that complies with the CC BY 4.0 license. Please check copyright information on all replacement figures and update the figure caption with source information. If applicable, please specify in the figure caption text when a figure is similar but not identical to the original image and is therefore for illustrative purposes only.

5.) Please include captions for your Supporting Information files at the end of your manuscript, and update any in-text citations to match accordingly. Please see our Supporting Information guidelines for more information: http://journals.plos.org/plosone/s/supporting-information.

Reviewers' comments:

Reviewer's Responses to Questions

**Comments to the Author**

1. Is the manuscript technically sound, and do the data support the conclusions?

Reviewer #1: Yes

Reviewer #2: No

2. Has the statistical analysis been performed appropriately and rigorously? 

Reviewer #1: No

Reviewer #2: Yes

3. Have the authors made all data underlying the findings in their manuscript fully available?

Reviewer #1: Yes

Reviewer #2: Yes

4. Is the manuscript presented in an intelligible fashion and written in standard English?

Reviewer #1: No

Reviewer #2: Yes

5. Review Comments to the Author

Reviewer #1: This study explores the impact of temperature effect on hospitalization of psychosis patients with analyzing the influence of different environmental and meteorological factors regarding different gender, ages, period, and cold/hot season on hospital admissions of various mental disorders. It especially takes temperate zone of Europe for high temperature effects research and is relatively interesting topic worthy of discussion. The followings are review recommendations:

Review Recommendations:

1. Studies showed that the health effect of temperature often presents a J or U-shaped curve. There are not many literatures and discussions about the impact of high temperature on possible mental disorders and possible pathogenic mechanisms. The high temperature circumstances in other tropical and subtropical zone are even more obvious than the studied area. It is not complete if only discussing on the effects and impacts of climate change, especially in the Introduction section. What is its uniqueness and importance?

2. The design of this research is a hospital-based study, which needs to further explain whether the inpatients with mental disorders have the problem of repeated medical hospitalization. Repeated medical hospitalization of mental disorders affected by environmental factors or seasonality are relatively common. Are these calculated as independent cases? It needs to verify the rationality.

3. The research period lasted up to 45 years. Has the diagnostic criteria ever been revised and adjusted? The inpatients with mental disorders suddenly increased since 1995 in Figure 2. Is the drastic change caused by the conversion in ICD code? Moreover, it does not clearly explain the conversion between Icd9 and icd10 of different disease classification codes. What is the classification standard between different mental diseases?

4. The mild and severe mental disorders by classification have different hospitalization requirements. This research classified only by different disease codes does not consider the severity of illness and length of hospitalization. In addition, one type of mental disorder often has other psychiatric comorbidities and characteristics. How to diagnose and classify? Therefore, it is necessary to discuss the definition and type of hospitalization and the severity of different diseases.

5. What is the reason and purpose to adopt the population-weighted daily mean temperature estimation method for this research? Is it used to estimate the temperature of the patient’s residence, or is it the average temperature of neighboring hospital taken as the case’s exposure temperature? How are the parameters of other environmental and weather factors further analyzed and used?

6. It doesn’t explain clearly about the statistical analysis method. How to prove that the temperature effect is linear? The temperature effect usually uses DLNM as the analysis method. Why not adopt it? The method of model sensitivity analysis is also not explained.

7. The analysis model and its included variables are not clearly presented. Figure S3 shows that the highest and significant risk falls between lag0 and lag1. Why use lag3 to analyze the total data? Since DLM is adopted, why not discuss the different delay times of lag0, lag1, lag2 ?

8. This study adopts the percentile method for high temperature. The definition of heat wave and its duration need to be supplemented and clearly defined. What is the definition of Extreme heat events in Table 2 and Extreme temperature events in Table S1? Why are the temperatures higher than the 90th and 95th percentile for lasting two days both different between the two tables? How to define the events?

9. Some graphs (fig2, fig3) have wrong texts and illustrations. Whether the text descriptions are significant or insignificant to statistics, its representative meaning is not clear. The explanation of statistical results must be more precise.

10. Line 182 - Is the data period 1973/1/1 to 2017/12/31? Or until 2018/12/31? Line 182 - Is the total number of people 89,869 or 93,655? Line 184 - Is the ratio of elders over 65 in the data 16% or 14%? The total number of people diagnosed with diseases is 84,397 in Table 1, and 5,472 people among them have no data due to blank ICD code. Why are these people included in this study?

11. Figure S4 shows significant risks only in temperature between 0°C and minus 10°C. Why is the main axis of the whole study discussing the effect of high temperature? Why not discuss the effect of low temperature?

Reviewer #2: This is an observational study to investigate ambient temperature and mental health hospitalizations in Bern, Switzerland for 45 years. The conclusions of this study indicate that increasing temperatures could negatively affect mental status in psychiatric patients (4.0% for every 10° C). Although the topic is interesting, some points should be addressed.

1. The authors documented that larger risks were found for hospitalizations related developmental disorders and schizophrenia. However, I think temperature may not be influence on aggravation of symptoms of both disorders. Many confounders should be carefully evaluated to make such a conclusion. I think the mental system of Bern and family care may be confounders between temperature and mental health hospitalization I think my country shows no association between temperature and psychiatric admissions.

2. Seasonal variation and sunlight should be carefully evaluated. Both may influence on aggravation of mood symptoms in depression and bipolar disorders.

3. High temperature may interfere outpatient follow-up, so their symptoms may be aggravated. The authors should clarify their visit in outpatient clinics in summer seasons.

6. PLOS authors have the option to publish the peer review history of their article (what does this mean?). If published, this will include your full peer review and any attached files.

Reviewer #1: No

Reviewer #2: No

---

## [Author Response · Author response to Decision Letter 0]

6 Mar 2021

General comments on the review

We would like to thank the reviewers for their constructive and useful feedback which have greatly contributed to improve the quality of our work. We have carefully addressed all the points of the review in the current version of the manuscript and we have provided a detailed answer to each of the queries below. 

Comments from the editorial board

1.) Please ensure that your manuscript meets PLOS ONE's style requirements, including those for file naming. 

We confirm that our manuscript meets PLOS ONE's style requirements.

2.) Thank you for stating the following in the Financial Disclosure section: 'The authors received no specific funding for this work.' We note that one or more of the authors are employed by a commercial company: Privatclinic Meiringen, Switzerland.

2.1. Please provide an amended Funding Statement declaring this commercial affiliation, as well as a statement regarding the Role of Funders in your study. If the funding organization did not play a role in the study design, data collection and analysis, decision to publish, or preparation of the manuscript and only provided financial support in the form of authors' salaries and/or research materials, please review your statements relating to the author contributions, and ensure you have specifically and accurately indicated the role(s) that these authors had in your study. You can update author roles in the Author Contributions section of the online submission form.

Please also include the following statement within your amended Funding Statement: “The funder provided support in the form of salaries for authors [insert relevant initials], but did not have any additional role in the study design, data collection and analysis, decision to publish, or preparation of the manuscript. The specific roles of these authors are articulated in the ‘author contributions’ section.”

2.2. Please also provide an updated Competing Interests Statement declaring this commercial affiliation along with any other relevant declarations relating to employment, consultancy, patents, products in development, or marketed products, etc. 

Within your Competing Interests Statement, please confirm that this commercial affiliation does not alter your adherence to all PLOS ONE policies on sharing data and materials by including the following statement: "This does not alter our adherence to PLOS ONE policies on sharing data and materials.” (as detailed online in our guide for authors http://protect-eu.mimecast.com/s/1IrdCvZ14tWxQvVhAvgtt?domain=journals.plos.org) . If this adherence statement is not accurate and there are restrictions on sharing of data and/or materials, please state these. Please note that we cannot proceed with consideration of your article until this information has been declared.

We have added the requested information to the Funding Statement in the manuscript:

"Privatclinic Meiringen (Switzerland) provided support in the form of salaries for the author TM but did not have any additional role in the study design, data collection and analysis, decision to publish, or preparation of the manuscript. The specific roles of these authors are articulated in the ‘author contributions’ section."

We have updated the Competing Interests and added the requested funding formation:

"The author TM is employed by Privatclinic Meiringen, Switzerland and the author AT is employed by the European Commission. This does not alter our adherence to PLOS ONE policies on sharing data and materials. The views here expressed are those of the authors and do not necessarily reflect an official position of the European Commission. The authors have declared that no competing interests exist."

3.) We note that you have indicated that data from this study are available upon request. PLOS only allows data to be available upon request if there are legal or ethical restrictions on sharing data publicly. For information on unacceptable data access restrictions, please see http://journals.plos.org/plosone/s/data-availability#loc-unacceptable-data-access-restrictions.

We will make the data publicly available online on the institutional repository of the University of Bern. 

4.) We note that Figure 1 in your submission contain map images which may be copyrighted. All PLOS content is published under the Creative Commons Attribution License (CC BY 4.0), which means that the manuscript [...]

4.1. You may seek permission from the original copyright holder of Figure1 to publish the content specifically under the CC BY 4.0 license. 

We ensure that there are no copyright issues with Figure 1, as it was originally created by one of the authors. The two maps presented in this figure were originally created in R (version 3.4.4, R Development Core Team). R is available as free software under the terms of the Free Software Foundation's GNU General Public License in source code form. Additionally, the layout of the maps was built in Adobe Illustrator by one of the authors. 

5.) Please include captions for your Supporting Information files at the end of your manuscript, and update any in-text citations to match accordingly. Please see our Supporting Information guidelines for more information: http://protect-eu.mimecast.com/s/v4tZCO8KBCAnW3QFBOzbn?domain=journals.plos.org.

We included captions for the Supporting Information files at the end of the manuscript and updated the in-text citations. 

Reviewer 1

This study explores the impact of temperature effect on hospitalization of psychosis patients with analyzing the influence of different environmental and meteorological factors regarding different gender, ages, period, and cold/hot season on hospital admissions of various mental disorders. It especially takes temperate zone of Europe for high temperature effects research and is relatively interesting topic worthy of discussion. The followings are review recommendations:

We thank the reviewer for acknowledging the value of our study.

Review Recommendations:

1. Studies showed that the health effect of temperature often presents a J or U-shaped curve. There are not many literatures and discussions about the impact of high temperature on possible mental disorders and possible pathogenic mechanisms. The high temperature circumstances in other tropical and subtropical zone are even more obvious than the studied area. It is not complete if only discussing on the effects and impacts of climate change, especially in the Introduction section. What is its uniqueness and importance?

Thanks to the reviewer for the useful comment. We have amended the text in the introduction and discussion to highligh the relevance and the uniqueness of our study. With these edits (see tracked changes version), we seek to highlight that evidence on the topic is currently scarce, in particular in Europe or temperate regions such as Switzerland. 

Our results show that psychiatric patients are more vulnerable to high temperature, which is important to emphasize for policy-makers as it is possible then that this burden would probably amplify under current climate change projections if no adequate public health measures are put inplaced.

2. The design of this research is a hospital-based study, which needs to further explain whether the inpatients with mental disorders have the problem of repeated medical hospitalization. Repeated medical hospitalization of mental disorders affected by environmental factors or seasonality are relatively common. Are these calculated as independent cases? It needs to verify the rationality.

Thanks to the reviewer for raising this point. Unfortunately, we did not have information on the identity of the patient nor on whether the case was recurrent or not. We have noted this limitation in the text (384-388). We agree with the reviewer that a sensitivity analysis accounting for recurrent hospitalizations in the statistical approach (i.e. mix model approach assuming non-independence) would have been more appropriate. However, it should be noted that assuming independence of the cases, when these are actually non-independent, would have only affected the precision of the estimates but not the magnitude or direction of the observed effect. In our analysis we also assume that the role of temperature in recurrent or non-recurrent events is the same. However, we also agree with the reviewer it would have been interesting to explore potential differential effects. 

3. The research period lasted up to 45 years. Has the diagnostic criteria ever been revised and adjusted? The inpatients with mental disorders suddenly increased since 1995 in Figure 2. Is the drastic change caused by the conversion in ICD code? Moreover, it does not clearly explain the conversion between Icd9 and icd10 of different disease classification codes. What is the classification standard between different mental diseases?

We thank the reviewer for pointing out this important issue. We have included additional text in the method section to clarify this issue. As acknowledged now in the manuscript, it is possible that the diagnostic criteria could have changed during the study period causing potential artificial steps or anomalous patterns in the temporal trend of the daily number of hospitalizations (as the one observed in 1995). In addition to this, we already stated in the text that there was a change in codification criteria from ICD 9 to ICD 10 in 1994. We have included an additional sentence in the methods section that better explains how the ICD conversion was performed (lines 101-103). As we explain now in the text (lines 143-145), these changes in codification/diagnosis could have altered the long-term trends of the outcome, but these could have not introduced any bias in our estimations, as these alterations in the temporal trends are controlled for by design (i.e. aggregated case-crossover) (lines 139-145, 359-362).

4. The mild and severe mental disorders by classification have different hospitalization requirements. This research classified only by different disease codes does not consider the severity of illness and length of hospitalization. In addition, one type of mental disorder often has other psychiatric comorbidities and characteristics. How to diagnose and classify? Therefore, it is necessary to discuss the definition and type of hospitalization and the severity of different diseases.

We thank the reviewer for the interesting comment. Unfortunately, our database had very limited individual information, and data on the severity of the illness or potential comorbidities was missing. However, we agree with the reviewer that evidence on potential differential effects by severity/comorbidities would have been very valuable. We extended the text on this limitation in the discussion part (lines 373-377).

5. What is the reason and purpose to adopt the population-weighted daily mean temperature estimation method for this research? Is it used to estimate the temperature of the patient’s residence, or is it the average temperature of neighboring hospital taken as the case’s exposure temperature? How are the parameters of other environmental and weather factors further analyzed and used?

Thanks for raising this issue, we acknowledge that possibly this section of the analysis was not described in enough detail. We have amended the text in the method section with new information to clarify the rationale of the approach. We decided to use population-weighted spatially-resolved temperature data, instead of the usual weather station data, because we believe that it would be a better representation of the average exposure of the population in the catchment area (lines 114-118). This is the conclusion of a study by De Schrijver et al. (currently under review in GeoHealth journal), performed by some of the authors of this manuscript . Population-weighted temperature was estimated in the following way: 1) we used the ratio between the population residing in the corresponding grid cell and the residing total population in the catchment area to compute the weights of temperature cells. 2) Then, we computed the daily mean temperature as a weighted average across the temperature in these grid cells using the corresponding weight. The main reason for doing this is that the distribution of the population across the catchment area of the hospital is highly heterogeneous due to the irregular orography of the region. Thus, with this approach (population weighting), we overcome potential exposure misclassification in regions where the exposure of the population is not fully captured by either unweighted estimates or weather station data. It should be noted that this is an ecological study (i.e. exposure and outcome are average across the study area) and we did not have information on the residence of the patients. 

For other environmental factors (weather, except temperature, and air pollutants), we used the non-weighted population data provided by the stations as spatially-resolved datasets were not available. This is alreayd stated in the manuscript (lines 121-124).

6. It doesn’t explain clearly about the statistical analysis method. How to prove that the temperature effect is linear? The temperature effect usually uses DLNM as the analysis method. Why not adopt it? The method of model sensitivity analysis is also not explained.

Thanks for pointing out this issue, we believe that the description of the method in the initial version of the manuscript was not accurate enough. We have modified the text to clarify this issue. 

We actually explored potential non-linearities of the association using DLNMs (as suggested by the reviewer). However, according to AIC criteria, the model assuming a linear relationship provided a better fit - the reason why we used the usual distributed lag models (DLMs), not DLNMs in the main analysis. 

We describe in the suppl file the different models used in the sensitivity analyses (Table S4) and we show the resulting curve from the model with DLNMs which actually turns to be close to be linear (Fig S4) - reassuring our choice on the linear function to model the association.

7. The analysis model and its included variables are not clearly presented. Figure S3 shows that the highest and significant risk falls between lag0 and lag1. Why use lag3 to analyze the total data? Since DLM is adopted, why not discuss the different delay times of lag0, lag1, lag2 ?

We thank the reviewer for the suggestion. Although the largest RR are observed between lag0 and lag1, we decided to account until lag3 and provide an overall estimate that represents cumulative association or risk from lag0 to lag3. We decided to not focus on the lag-response pattern of the association because it would add more complexity to the analysis. However, we believe that the reported estimate (cumulative exposure-response until lag 3) is an appropriate representation of the association of interest. It should be noted that in the sensitivity analyses, we showed that extending the lag to 7 days did not lead to significant changes in the outcome.

8. This study adopts the percentile method for high temperature. The definition of heat wave and its duration need to be supplemented and clearly defined. What is the definition of Extreme heat events in Table 2 and Extreme temperature events in Table S1? Why are the temperatures higher than the 90th and 95th percentile for lasting two days both different between the two tables? How to define the events?

We apologize for not providing clear definitions for the extreme heat events. We have corrected the text in the methods part and Table 1 to clarify the different criteria used for each type of heatwave. There are no standard definitions for heatwaves consistently used in literature. We decided to incorporate in the analysis a set of definitions that have been widely used in the previous analysis (Guo Y, Gasparrini A, Armstrong BG, Tawatsupa B, Tobias A, Lavigne E, et al. Heat Wave and Mortality: A Multicountry, Multicommunity Study. Environmental health perspectives. 2017;125(8):087006). We have also added this information in the manuscript. We also apologize for the mistakes in Table S1 - as now clarified, we are presenting descriptive data for the heatwaves of two or three days above the 92.5th and 97.5 percentile of temperature. 

9. Some graphs (fig2, fig3) have wrong texts and illustrations. Whether the text descriptions are significant or insignificant to statistics, its representative meaning is not clear. The explanation of statistical results must be more precise.

We have added the following statement in the figure legend, according to the reviewer's suggestion: "Null hypothesis is that there is no association (RR=1), thus one can reject the null hypothesis when 95% confidence interval does not include 1".

10. Line 182 - Is the data period 1973/1/1 to 2017/12/31? Or until 2018/12/31? Line 182 - Is the total number of people 89,869 or 93,655? Line 184 - Is the ratio of elders over 65 in the data 16% or 14%? The total number of people diagnosed with diseases is 84,397 in Table 1, and 5,472 people among them have no data due to blank ICD code. Why are these people included in this study?

We apologized for these mistakes in the text. The total number of patients is 88,996 and the ratio of elders over 65 is 15%. We have corrected this information in the manuscript. The study period dates are correctly provided (from 1973/1/1 to 2017/12/31). As noted by the reviewer, there were 5,447 patients (6.5%) for which data on the diagnosis was not available. However, we decided to not keep them in the main analysis as data on the other variables was available (age, sex) and to maximize the statistical power. 

11. Figure S4 shows significant risks only in temperature between 0°C and minus 10°C. Why is the main axis of the whole study discussing the effect of high temperature? Why not discuss the effect of low temperature?

As explained in comment 6, when applying a non-linear function the resulting curve (shown in Figure S4 bottom) was close to linear, which supported our choice of using a linear function in the main analysis. This would imply that the risk is the same in warm and cold ranges, and with the same direction, magnitude and precision. 

Reviewer #2

This is an observational study to investigate ambient temperature and mental health hospitalizations in Bern, Switzerland for 45 years. The conclusions of this study indicate that increasing temperatures could negatively affect mental status in psychiatric patients (4.0% for every 10° C). Although the topic is interesting, some points should be addressed.

1. The authors documented that larger risks were found for hospitalizations related developmental disorders and schizophrenia. However, I think temperature may not be influence on aggravation of symptoms of both disorders. Many confounders should be carefully evaluated to make such a conclusion. I think the mental system of Bern and family care may be confounders between temperature and mental health hospitalization I think my country shows no association between temperature and psychiatric admissions.

We thank the reviewer for his/her comment. As mentioned in the discussion there are several other studies suggesting that in schizophrenic patients hospitalizations may increase and symptoms may be aggravated due to heat (1. Gupta S, Murray RM. The relationship of environmental temperature to the incidence and outcome of schizophrenia. The British journal of psychiatry : the journal of mental science. 1992;160:788-92. 2. Hermesh H, Shiloh R, Epstein Y, Manaim H, Weizman A, Munitz H. Heat intolerance in patients with chronic schizophrenia maintained with antipsychotic drugs. The American journal of psychiatry. 2000;157(8):1327-9. 3. Shiloh R, Shapira A, Potchter O, Hermesh H, Popper M, Weizman A. Effects of climate on admission rates of schizophrenia patients to psychiatric hospitals. European psychiatry : the journal of the Association of European Psychiatrists. 2005;20(1):61-4. 4. Shiloh R, Munitz H, Stryjer R, Weizman A. A significant correlation between ward temperature and the severity of symptoms in schizophrenia inpatients--a longitudinal study. European neuropsychopharmacology : the journal of the European College of Neuropsychopharmacology. 2007;17(6-7):478-82. 5. Hansen A, Bi P, Nitschke M, Ryan P, Pisaniello D, Tucker G. The effect of heat waves on mental health in a temperate Australian city. Environmental health perspectives. 2008;116(10):1369-75). Also, another study suggests that heat leads to aggravated symptoms also in patients with developmental disorders (Hansen A, Bi P, Nitschke M, Ryan P, Pisaniello D, Tucker G. The effect of heat waves on mental health in a temperate Australian city. Environmental health perspectives. 2008;116(10):1369-75.). We believe that our estimates are not confounded by other individual factors, because this is accounted for by design, as mentioned now in the text (lines 359-362). 

2. Seasonal variation and sunlight should be carefully evaluated. Both may influence on aggravation of mood symptoms in depression and bipolar disorders.

We agree with the reviewer that seasonality or other weather factors like sunlight may act as possible confounders of the association. This is the reason why we first controlled for seasonality using the matching strategy (see lines 139-143). We believe that our analysis properly accounts for seasonal patterns. Additionally, in the sensitivity analyses (S1 Fig), we found that sunlight was not associated with daily hospitalizations, thus it should not be considered by definition a confounder of the temperature-hospitalization association. 

3. High temperature may interfere outpatient follow-up, so their symptoms may be aggravated. The authors should clarify their visit in outpatient clinics in summer seasons.

Thanks to the reviewer for the suggestion. We agree that assessing the effect of temperature on outpatient visits would provide additional insights. Unfortunately, these data were not available. We mention this limitation in the discussion section (lines 373-377).

---

## [Decision Letter · Decision Letter 1]

16 Aug 2021

PONE-D-20-33096R1

Ambient temperature and mental health hospitalizations in Bern, Switzerland: A 45-year time-series study

PLOS ONE

Dear Dr. Ana,

Thank you for submitting your manuscript to PLOS ONE. After careful consideration, we feel that it has merit but does not fully meet PLOS ONE’s publication criteria as it currently stands. Therefore, we invite you to submit a revised version of the manuscript that addresses the points raised during the review process.

The potential biological mechanisms regarding the effects of ambient temperature on mental health hospitalizations should be discussed in detail in the DISCUSSION section with enough references cited. Limitations of this study should also stated with details in the DISCUSSION section.

We look forward to receiving your revised manuscript.

Kind regards,

Mingqing Xu

Academic Editor

PLOS ONE

Journal Requirements:

Additional Editor Comments (if provided):

The potential biological mechanisms regarding the effects of ambient temperature on mental health hospitalizations should be discussed in detail in the DISCUSSION section with enough references cited. Limitations of this study should also stated with details in the DISCUSSION section.

Reviewers' comments:

Reviewer's Responses to Questions

**Comments to the Author**

1. If the authors have adequately addressed your comments raised in a previous round of review and you feel that this manuscript is now acceptable for publication, you may indicate that here to bypass the “Comments to the Author” section, enter your conflict of interest statement in the “Confidential to Editor” section, and submit your "Accept" recommendation.

Reviewer #1: (No Response)

2. Is the manuscript technically sound, and do the data support the conclusions?

Reviewer #1: Yes

3. Has the statistical analysis been performed appropriately and rigorously? 

Reviewer #1: Yes

4. Have the authors made all data underlying the findings in their manuscript fully available?

Reviewer #1: Yes

5. Is the manuscript presented in an intelligible fashion and written in standard English?

Reviewer #1: Yes

6. Review Comments to the Author

Reviewer #1: 1. This study used case-crossover design to compare different exposure periods for the same population. It needs a more detailed explanation for the hospitalization difference (figure2) caused by the conversion of icd9/icd10 and also for the possibility of underestimation and overestimation of its influence (association) resulted from the deviation.

2. The explanation of patients' repeated hospitalization and disease severity is too brief.

3. It should discuss more on the mechanism or the reason why developmental disorders and schizophrenia are affected by temperatures in the discussion section.

4. What are the possible reasons of that extremely high temperature (heatwave) are not significant? Please add in discussion section.

5. The author stated that the risk is the same in warm and cold ranges, and with the same direction, magnitude and precision. (Response to Review1 Recommendations) How to come out this conclusion?

7. PLOS authors have the option to publish the peer review history of their article (what does this mean?). If published, this will include your full peer review and any attached files.

Reviewer #1: No

---

## [Author Response · Author response to Decision Letter 1]

23 Sep 2021

General comments on the review

We would like to thank once again the reviewers for their constructive comments and suggestions to improve the quality of our manuscript. We have carefully addressed all the points of the review in the current version of the manuscript and we have provided a detailed answer to each of the queries below. 

Comments from the editorial board

The potential biological mechanisms regarding the effects of ambient temperature on mental health hospitalizations should be discussed in detail in the DISCUSSION section with enough references cited. Limitations of this study should also stated with details in the DISCUSSION section. 

We have included additional information in the discussion section on the biological mechanism explaining the effect of ambient temperature on mental health and, in particular, for schizophrenia and developmental disorders for which larger estimates were found. 

Additionally, as suggested, we have extended the section on the study limitations in the discussion section. 

The following references have been added to the manuscript: 

17. IPCC, 2021: Summary for Policymakers. In: Climate Change 2021: The Physical Science Basis. Contribution of Working Group I to the Sixth Assessment Report of the Intergovernmental Panel on Climate Change [MassonDelmotte, V., P. Zhai, et al. (eds.)]. Cambridge University Press. In Press.

21. Butler R. The ICD-10 General Equivalence Mappings. Bridging the translation gap from ICD-9. Journal of AHIMA. 2007;78(9):84-5.

35. Ragettli MS, Röösli M. [Heat-health action plans to prevent heat-related deaths-experiences from Switzerland]. Bundesgesundheitsblatt, Gesundheitsforschung, Gesundheitsschutz. 2019;62(5):605-11.

42. Cianconi P, Betrò S, Janiri L. The Impact of Climate Change on Mental Health: A Systematic Descriptive Review. Front Psychiatry. 2020;11(74).

43. Maes M, Scharpé S, Verkerk R, D'Hondt P, Peeters D, Cosyns P, et al. Seasonal variation in plasma L-tryptophan availability in healthy volunteers. Relationships to violent suicide occurrence. Archives of general psychiatry. 1995;52(11):937-46.

48. Dunn W. The impact of sensory processing abilities on the daily lives of young children and their families: A conceptual model. Infants and young children. 1997;9:23-35.

49. Dunn W. The sensations of everyday life: empirical, theoretical, and pragmatic considerations. The American journal of occupational therapy : official publication of the American Occupational Therapy Association. 2001;55(6):608-20.

50. Miller LJ, Anzalone ME, Lane SJ, Cermak SA, Osten ET. Concept evolution in sensory integration: a proposed nosology for diagnosis. The American journal of occupational therapy : official publication of the American Occupational Therapy Association. 2007;61(2):135-40.

51. Tada K, Tateda H, Arashima S, Sakai K, Kitagawa T, Aoki K, et al. Follow-up study of a nation-wide neonatal metabolic screening program in Japan. A collaborative study group of neonatal screening for inborn errors of metabolism in Japan. European journal of pediatrics. 1984;142(3):204-7.

52. Williams ZJ, Failla MD, Davis SL, Heflin BH, Okitondo CD, Moore DJ, et al. Thermal Perceptual Thresholds are typical in Autism Spectrum Disorder but Strongly Related to Intra-individual Response Variability. Scientific Reports. 2019;9(1):12595.

53. Bolton M, Blumberg W, Ault L, Mogil H, Hanes S. Initial Evidence for Increased Weather Salience in Autism Spectrum Conditions. Weather, Climate, and Society. 2020;12.

54. Scahill L, Lombroso PJ, Mack G, Van Wattum PJ, Zhang H, Vitale A, et al. Thermal sensitivity in Tourette syndrome: preliminary report. Perceptual and motor skills. 2001;92(2):419-32.

55. Park RJ, Goodman J, Hurwitz M, Smith J. Heat and Learning. American Economic Journal: Economic Policy. 2020;12(2):306-39.

61. Bouchama A, Dehbi M, Mohamed G, Matthies F, Shoukri M, Menne B. Prognostic factors in heat wave related deaths: a meta-analysis. Archives of internal medicine. 2007;167(20):2170-6.

62. Davido A, Patzak A, Dart T, Sadier MP, Méraud P, Masmoudi R, et al. Risk factors for heat related death during the August 2003 heat wave in Paris, France, in patients evaluated at the emergency department of the Hôpital Européen Georges Pompidou. Emerg Med J. 2006;23(7):515-8.

Reviewer 1

Review Recommendations:

1. This study used case-crossover design to compare different exposure periods for the same population. It needs a more detailed explanation for the hospitalization difference (figure2) caused by the conversion of icd9/icd10 and also for the possibility of underestimation and overestimation of its influence (association) resulted from the deviation.

Thanks to the reviewer for the useful comment. We investigated further this issue and asked the hospital for further details about this increase in hospitalizations cases. Even though in this period there was a change in the classification of the disease, they considered that this was not the main reason for the increased number of mental health hospitalizations (contrary to what we stated in the previous review). In 1994, the management of the hospital took the administrative decision to shorten the length of stay for hospitalized patients. This could have led to an increase in incident hospitalizations cases. However, we would like to point that even though these changes in the length of stay could have altered the long-term trends of the outcome, this could have not introduced any bias in our estimations. As now clarified in the methods section, in the case-crossover design each day is compared to control days selected based on the day of the week, month and year. That is, for example, the exposure and outcome in the second Monday of January 1993 are compared with the exposure and outcome in all the other Mondays of January 1993. Thus, estimates reflect variations in the exposure-outcome happening on the short-term (within a month) and any temporal patterns occurring at longer time scales beyond that time frame are removed by design. This was clarified in the manuscript (lines 216-222: “Deviation from monotonicity in the long-term behaviour and change points can be also observed, as for (e.g. example the jump steep increase from in 1996 onwards). These, which could be possibly attributed to changes in diagnostic criteria or administrative decisions (e.g. reduction in the length of the stay which would allow for the admission of more patients). However, as mentioned in the method section, factors altering long-term temporal patterns would not introduce bias in the association estimates, as its potential confounding effect is removed by design”, lines 423-428). 

2. The explanation of patients' repeated hospitalization and disease severity is too brief.

Thanks to the reviewer for raising this point. We have expanded the explanation for repeated hospitalizations and disease severity in the discussion section (lines 416-420).

3. It should discuss more on the mechanism or the reason why developmental disorders and schizophrenia are affected by temperatures in the discussion section.

We thank the reviewer for pointing out this important issue. We have included additional text in the discussion section on the mechanism of how ambient temperature affects schizophrenia and different developmental disorders (autism spectrum disorders, attention-deficit/hyperactivity disorder, tic syndromes and learning disabilities (lines 341-371).

4. What are the possible reasons of that extremely high temperature (heatwave) are not significant? Please add in discussion section.

We thank the reviewer for raising this issue. The non-significant effect of heatwaves can be explained by the low statistical power for this type of event in the study. Even though, heatwave occurrences have increased steadily in Switzerland in the last years, they still remain a rare event (with an average of 14-35 heatwave days per year (depending on the definition of heatwaves) (Table 2). We have added this explanation in the text (lines 303-305).

5. The author stated that the risk is the same in warm and cold ranges, and with the same direction, magnitude and precision. (Response to Review1 Recommendations) How to come out this conclusion?

Thanks for the comment, and we apologize if the statement was not clear in the previous review. As described in the manuscript, the association between temperature and the risk of hospitalizations was linear, thus the magnitude of the association (expressed as an increase in risk per 1C) is constant across the whole range of temperatures. Thus, according to our findings the risk of mental hospitalizations increased by 4% per 10C increase, regardless it is an increase from 15 to 25C, or -5 to 5C (i.e. warm and cold ranges).

---

## [Editor Report · Decision Letter 2]

24 Sep 2021

Ambient temperature and mental health hospitalizations in Bern, Switzerland: A 45-year time-series study

PONE-D-20-33096R2

Dear Dr. Ana Maria Vicedo-Cabrera,

We’re pleased to inform you that your manuscript has been judged scientifically suitable for publication and will be formally accepted for publication once it meets all outstanding technical requirements.

Kind regards,

Mingqing Xu

Academic Editor

PLOS ONE

Additional Editor Comments (optional):

It can be accepted for publication now.
---

## [Editor Report · Acceptance letter]

1 Oct 2021

PONE-D-20-33096R2 

Ambient temperature and mental health hospitalizations in Bern, Switzerland: A 45-year time-series study 

Dear Dr. Vicedo-Cabrera:

I'm pleased to inform you that your manuscript has been deemed suitable for publication in PLOS ONE. Congratulations! Your manuscript is now with our production department. 

Kind regards, 

on behalf of

Dr. Mingqing Xu 

Academic Editor

PLOS ONE